# Rethinking Deep Neural Network Ownership Verification: Embedding Passports to Defeat Ambiguity Attacks

**Lixin Fan**[1]  **Kam Woh Ng**[2]  **Chee Seng Chan**[2]
[1]WeBank AI Lab, Shenzhen, China
[2]Center of Image and Signal Processing, Faculty of Comp. Sci. and Info., Tech.
University of Malaya, Kuala Lumpur, Malaysia
{lixinfan@webank.com;kamwoh@siswa.um.edu.my;cs.chan@um.edu.my}

## Abstract

With substantial amount of time, resources and human (team) efforts invested to explore and develop successful deep neural networks (DNN), there emerges an urgent need to protect these inventions from being illegally copied, redistributed, or abused without respecting the intellectual properties of legitimate owners. Following recent progresses along this line, we investigate a number of watermark-based DNN ownership verification methods in the face of ambiguity attacks, which aim to cast doubts on the ownership verification by forging counterfeit watermarks. It is shown that ambiguity attacks pose serious threats to existing DNN watermarking methods. As remedies to the above-mentioned loophole, this paper proposes novel *passport*-based DNN ownership verification schemes which are both *robust to network modifications* and *resilient to ambiguity attacks*. The gist of embedding digital passports is to design and train DNN models in a way such that, the DNN inference performance of an original task will be significantly *deteriorated due to forged passports*. In other words, genuine passports are not only verified by looking for the predefined signatures, but also reasserted by the *unyielding DNN model inference performances*. Extensive experimental results justify the effectiveness of the proposed passport-based DNN ownership verification schemes. Code and models are available at `https://github.com/kamwoh/DeepIPR`

## 1 Introduction

With the rapid development of deep neural networks (DNN), Machine Learning as a Service (MLaaS) has emerged as a viable and lucrative business model. However, building a successful DNN is not a trivial task, which usually requires substantial investments on expertise, time and resources. As a result of this, there is an urgent need to protect invented DNN models from being illegally copied, redistributed or abused (i.e. intellectual property infringement). Recently, for instance, digital *watermarking* techniques have been adopted to provide such a protection, by embedding watermarks into DNN models during the training stage. Subsequently, ownerships of these inventions are verified by the detection of the embedded watermarks, which are supposed to be robust to multiple types of modifications such as model fine-tuning, model pruning and watermark overwriting [1–4].

In terms of deep learning methods to embed watermarks, existing approaches can be broadly categorized into two schools: a) the *feature-based* methods that embed designated watermarks into the DNN weights by imposing additional regularization terms [1, 3, 5]; and b) the *trigger-set* based methods that rely on adversarial training samples with specific labels (i.e. backdoor trigger sets) [2, 4]. Watermarks embedded with either of these methods have successfully demonstrated robustness

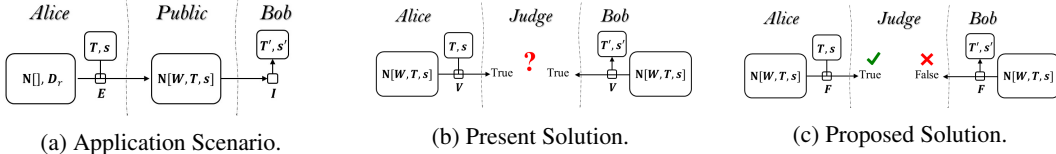

(a) Application Scenario.        (b) Present Solution.        (c) Proposed Solution.

Figure 1: DNN model ownership verification in the face of ambiguity attacks. **(a)**: Owner *Alice* uses an embedding process $E$ to train a DNN model with watermarks $(\mathbf{T}, \mathbf{s})$ and releases the model publicly available; Attacker *Bob* forges counterfeit watermarks $(\mathbf{T}', \mathbf{s}')$ with an invert process $I$; **(b)**: The ownership is in doubt since both the original and forged watermarks are detected by the verification process $V$ (Sect. 2.2); **(c)**: The ambiguity is resolved when our proposed passports are embedded and the network performances are evaluated in favour of the original passport by the fidelity evaluation process $F$ (See Definition 1 and Sect. 3.3).

against *removal attacks* which involve modifications of the DNN weights such as *fine-tuning* or *pruning*. However, our studies disclose the existence and effectiveness of *ambiguity attacks* which aim to cast doubt on the ownership verification by *forging additional watermarks* for DNN models in question (see Fig. 1). We also show that *it is always possible to reverse-engineer forged watermarks at minor computational cost* where the original training dataset is also not needed (Sect. 2).

As remedies to the above-mentioned loophole, this paper proposes a novel *passport*-based approach. There is a unique advantage of the proposed passports over traditional watermarks - i.e. the inference performance of a pre-trained DNN model will either *remain intact* given the presence of valid passports, or be *significantly deteriorated* due to either the modified or forged passports. In other words, we propose to *modulate the inference performances* of the DNN model depending on the presented passports, and by doing so, one can develop ownership verification schemes that are both *robust to removal attacks* and *resilient to ambiguity attacks* at once (Sect. 3).

The contributions of our work are threefold: i) we put forth a general formulation of DNN ownership verification schemes and, empirically, we show that existing DNN watermarking methods are vulnerable to ambiguity attacks; ii) we propose novel passport-based verification schemes and demonstrate with extensive experiment results that these schemes successfully defeat ambiguity attacks; iii) methodology-wise, the proposed modulation of DNN inference performance based on the presented passports (Eq. 4) plays an indispensable role in bringing the DNN model behaviours under control against adversarial attacks.

## 1.1 Related work

Uchida et. al [1] was probably the first work that proposed to embed watermarks into DNN models by imposing an additional *regularization term* on the weights parameters. [2, 6] proposed to embed watermarks in the classification labels of adversarial examples in a *trigger set*, so that the watermarks can be extracted remotely through a service API without the need to access the network weights (i.e. black-box setting). Also in both black-box and white box settings, [3, 5, 7] demonstrated how to embed watermarks (or fingerprints) that are robust to various types of attacks. In particular, it was shown that embedded watermarks are in general robust to *removal attacks* that modify network weights via fine-tuning or pruning. Watermark overwriting, on the other hand, is more problematic since it aims to simultaneously embed a new watermark and destroy the existing one. Although [5] demonstrated robustness against overwriting attack, it did not resolve the ambiguity resulted from the counterfeit watermark. Adi et al. [2] also discussed how to deal with an adversary who fine-tuned an already watermarked networks with new trigger set images. Nevertheless, [2] required the new set of images to be distinguishable from the true trigger set images. This requirement is however often unfulfilled in practice, and our experiment results show that none of existing watermarking methods are able to deal with ambiguity attacks explored in this paper (see Sect. 2).

In the context of digital image watermarking, [8, 9] have studied *ambiguity attacks* that aim to create an ambiguous situation in which a watermark is reverse-engineered from an already watermarked image, by taking advantage of the invertibility of forged watermarks [10]. It was argued that *robust watermarks do not necessarily imply the ability to establish ownership*, unless *non-invertible watermarking* schemes are employed (see Proposition 2 for our proposed solution).

## 2 Rethinking Deep Neural Network Ownership Verification

This section analyses and generalizes existing DNN watermarking methods in the face of ambiguity attacks. We must emphasize that the analysis mainly focuses on three aspects i.e. *fidelity, robustness* and *invertibility* of the ownership verification schemes, and we refer readers to representative previous work [1–4] for formulations and other desired features of the entire watermark-based intellectual property (IP) protection schemes, which are out of the scope of this paper.

### 2.1 Reformulation of DNN ownership verification schemes

Figure 1 summarizes the application scenarios of DNN model ownership verifications provided by the watermark based schemes. Inspired by [10], we also illustrate an ambiguous situation in which rightful ownerships cannot be uniquely resolved by the current watermarking schemes alone. This loophole is largely due to an intrinsic weakness of the watermark-based methods i.e. *invertibility*. Formally, the definition of DNN model ownership verification schemes is generalized as follows.

**Definition 1.** A DNN model ownership verification scheme is a tuple $\mathcal{V} = (E, F, V, I)$ of processes:

I) An *embedding* process $E\big(\mathbf{D}_r, \mathbf{T}, \mathbf{s}, \mathbb{N}[\cdot], L\big) = \mathbb{N}[\mathbf{W}, \mathbf{T}, \mathbf{s}]$, is a DNN learning process that takes *training data* $\mathbf{D}_r = \{\mathbf{X}_r, \mathbf{y}_r\}$ as inputs, and optionally, either trigger set data $\mathbf{T} = \{\mathbf{X}_T, \mathbf{y}_T\}$ or signature $\mathbf{s}$, and outputs the model $\mathbb{N}[\mathbf{W}, \mathbf{T}, \mathbf{s}]$ by minimizing a given loss $L$.

*Remark*: the DNN architectures are pre-determined by $\mathbb{N}[\cdot]$ and, after the DNN weights $\mathbf{W}$ are learned, either the trigger set $\mathbf{T}$ or signatures $\mathbf{s}$ will be embedded and can be verified by the verification process defined next[1].

II) A *fidelity evaluation* process $F\big(\mathbb{N}[\mathbf{W}, \cdot, \cdot], \mathbf{D}_t, \mathcal{M}_t, \epsilon_f\big) = \{\textit{True, False}\}$ is to evaluate whether or not the discrepancy is less than a predefined threshold i.e. $|\mathcal{M}(\mathbb{N}[\mathbf{W}, \cdot, \cdot], \mathbf{D}_t) - \mathcal{M}_t| \leq \epsilon_f$, in which $\mathcal{M}(\mathbb{N}[\mathbf{W}, \cdot, \cdot], \mathbf{D}_t)$ is the DNN inference performance tested against a set of *test data* $\mathbf{D}_t$ where $\mathcal{M}_t$ is the target inference performance.

*Remark*: it is often expected that a well-behaved embedding process will not introduce a significant inference performance change that is greater than a predefined threshold $\epsilon_f$. Nevertheless, this fidelity condition remains to be verified for DNN models under either removal attacks or ambiguity attacks.

III) A *verification* process $V(\mathbb{N}[\mathbf{W}, \cdot, \cdot], \mathbf{T}, \mathbf{s}, \epsilon_s) = \{\textit{True, False}\}$ checks whether or not the expected signature $\mathbf{s}$ or trigger set $\mathbf{T}$ is successfully verified for a given DNN model $\mathbb{N}[\mathbf{W}, \cdot, \cdot]$.

*Remark*: for feature-based schemes, $V$ involves the detection of embedded signatures $\mathbf{s} = \{\mathbf{P}, \mathbf{B}\}$ with a false detection rate that is lesser than a predefined threshold $\epsilon_s$. Specifically, the detection boils down to measure the distances $D_f(f_e(\mathbf{W}, \mathbf{P}), \mathbf{B})$ between target feature $\mathbf{B}$ and features extracted by a transformation function $f_e(\mathbf{W}, \mathbf{P})$ parameterized by $\mathbf{P}$.

*Remark*: for trigger-set based schemes, $V$ first invokes a DNN inference process that takes trigger set samples $\mathbf{T}_x$ as inputs, and then it checks whether the prediction $f(\mathbf{W}, \mathbf{X}_T)$ produces the designated labels $\mathbf{T}_y$ with a false detection rate that is lesser than a threshold $\epsilon_s$.

IV) An *invert* process $I(\mathbb{N}[\mathbf{W}, \mathbf{T}, \mathbf{s}]) = \mathbb{N}[\mathbf{W}, \mathbf{T}', \mathbf{s}']$ exists and constitutes a successful *ambiguity attack*, if

(a) a set of new trigger set $\mathbf{T}'$ and/or signature $\mathbf{s}'$ can be reverse-engineered for a given DNN model;

(b) the forged $\mathbf{T}', \mathbf{s}'$ can be successfully verified with respect to the given DNN weights $\mathbf{W}$ i.e. $V\big(I(\mathbb{N}[\mathbf{W}, \mathbf{T}, \mathbf{s}]), \mathbf{T}', \mathbf{s}', \epsilon_s\big) = \textit{True}$;

(c) the fidelity evaluation outcome $F\big(\mathbb{N}[\mathbf{W}, \cdot, \cdot], \mathbf{D}_t, \mathcal{M}_t, \epsilon_f\big)$ defined in Definition 1.II remains *True*.

*Remark*: this condition plays an indispensable role in designing the non-invertible verification schemes to defeat ambiguity attacks (see Sect. 3.3).

| | Feature based method [1] | | | Trigger-set based method [2] | | |
|---|---|---|---|---|---|---|
| | CIFAR10 | Real WM Det. | Fake WM Det. | CIFAR10 | Real WM Det. | Fake WM Det. |
| CIFAR100 | 64.25 (90.97) | 100 (100) | 100 (100) | 65.20 (91.03) | 25.00 (100) | 27.80 (100) |
| Caltech-101 | 74.08 (90.97) | 100 (100) | 100 (100) | 75.06 (91.03) | 43.60 (100) | 46.80 (100) |

Table 1: Detection of embedded watermark (in %) with two representative watermark-based DNN methods [1, 2], before and after DNN weights fine-tuning for transfer learning tasks. Top row denotes a DNN model trained with CIFAR10 and weights fine-tuned for CIFAR100; while bottom row denotes weight fine-tuned for Caltech-101. Accuracy outside bracket is the transferred task, while in-bracket is the original task. WM Det. denotes the detection accuracies of real and fake watermarks.

V) If at least one invert process exists for a DNN verification scheme $\mathcal{V}$, then the scheme is called an *invertible* scheme and denoted by $\mathcal{V}^I = (E, F, V, I \neq \emptyset)$; otherwise, the scheme is called *non-invertible* and denoted by $\mathcal{V}^\emptyset = (E, F, V, \emptyset)$.

The definition as such is abstract and can be instantiated by concrete implementations of processes and functions. For instance, the following combined loss function (Eq. 1) generalizes loss functions adopted by both the feature-based and trigger-set based watermarking methods:

$$L = L_c\big(f(\mathbf{W}, \mathbf{X}_r), \mathbf{y}_r\big) + \lambda^t L_c\big(f(\mathbf{W}, \mathbf{X}_T), \mathbf{y}_T\big) + \lambda^r R(\mathbf{W}, \mathbf{s}), \qquad (1)$$

in which $\lambda^t, \lambda^r$ are the relative weight hyper-parameters, $f(\mathbf{W}, \mathbf{X}_-)$ are the network predictions with inputs $\mathbf{X}_r$ or $\mathbf{X}_T$. $L_c$ is the loss function like *cross-entropy* that penalizes discrepancies between the predictions and the target labels $\mathbf{y}_r$ or $\mathbf{y}_T$. Signature $\mathbf{s} = \{\mathbf{P}, \mathbf{B}\}$ consists of passports $\mathbf{P}$ and signature string $\mathbf{B}$. The regularization terms could be either $R = L_c(\sigma(\mathbf{W}, \mathbf{P}), \mathbf{B})$ as in [1] or $R = MSE(\mathbf{B} - \mathbf{PW})$ as in [3].

It must be noted that, for those DNN models that will be used for classification tasks, their inference performance $\mathcal{M}(\mathbb{N}[\mathbf{W}, \cdot, \cdot], \mathbf{D}_t) = L_c\big(f(\mathbf{W}, \mathbf{X}_t), \mathbf{y}_t\big)$ tested against a dataset $\mathbf{D}_t = \{\mathbf{X}_t, \mathbf{y}_t\}$ is independent of either the embedded signature $\mathbf{s}$ or trigger set $\mathbf{T}$. It is this independence that induces an invertible process for existing watermark-based methods as described next.

**Proposition 1** (*Invertible process*). For a DNN ownership verification scheme $\mathcal{V}$ as in Definition 1, if the fidelity process $F()$ is independent of either the signature $\mathbf{s}$ or trigger set $\mathbf{T}$, then there always exists an invertible process $I()$ i.e. the scheme is invertible $\mathcal{V}^I = (E, F, V, I \neq \emptyset))$.

### 2.2 Watermarking in the face of ambiguity attacks

As proved by Proposition 1, one is able to construct forged watermarks for any already watermarked networks. We tested the performances of two representative DNN watermarking methods [1, 2], and Table 1 shows that counterfeit watermarks can be forged for the given DNN models with 100% detection rate, and 100% fake trigger set images can be reconstructed as well in the original task. Given that the detection accuracies for the forged trigger set is slightly better than the original trigger set after fine-tuning, the claim of the ownership is ambiguous and cannot be resolved by neither feature-based nor trigger-set based watermarking methods. Shockingly, the computational cost to forge counterfeit watermarks is quite minor where the forging required no more than 100 epochs to optimize, and worst still this is achieved without the need of original training data.

In summary, the ambiguity attacks against DNN watermarking methods are effective with minor computational and without the need of original training datasets. We ascribe this loophole to the crux that the loss of the original task, i.e. $L_c\big(f(\hat{\mathbf{W}}, \mathbf{X}_r), \mathbf{y}_r\big)$ is *independent* of the forged watermarks. We refer readers to our extended version [11] for an elaboration on the ambiguity attack method we adopted and more detailed experiment results. In the next section, we shall illustrate a solution to defeat the ambiguity attacks.

## 3 Embedding passports for DNN ownership verification

The main motivation of embedding digital passports is to design and train DNN models in a way such that, their inference performances of the original task (i.e. classification accuracy) will be significantly *deteriorated due to the forged signatures*. We shall illustrate next first how to implement the desired property by incorporating the so called *passport layers*, followed by different ownership protection schemes that exploit the embedded passports to effectively defeat ambiguity attacks.

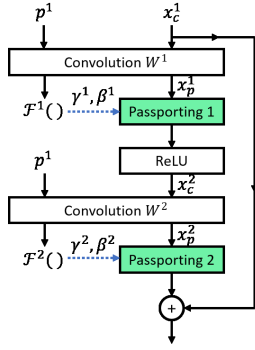

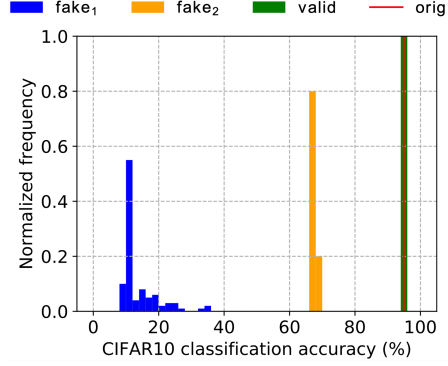

(a) An example in the *ResNet* layer that consists of the proposed passporting layers. $p^l = \{p_\gamma^l, p_\beta^l\}$ is the proposed *digital passports* where $\mathcal{F} = \text{Avg}(\mathbf{W}_p^l * \mathbf{P}_{\gamma,\beta}^l)$ is a passport function to compute the hidden parameters (i.e. $\gamma$ and $\beta$) given in Eq. (2).

(b) A comparison of CIFAR10 classification accuracies given the original DNN, proposed DNN with valid passports, proposed DNN with randomly generated passports ($fake_1$), and proposed DNN with reverse-engineered passports ($fake_2$).

Figure 2: (a) Passport layers in ResNet architecture and (b) Classification accuracies modulated by different passports in CIFAR10, e.g. given counterfeit passports, the DNN models performance will be deteriorated instantaneously to fend off illegal usage.

## 3.1 Passport layers

In order to control the DNN model functionalities by the embedded digital signatures i.e. *passports*, we proposed to append after a convolution layer a *passport layer*, whose scale factor $\gamma$ and bias shift term $\beta$ are dependent on both the convolution kernels $\mathbf{W}_p$ and the designated passport $\mathbf{P}$ as follows:

$$\mathbf{O}^l(\mathbf{X}_p) = \gamma^l \mathbf{X}_p^l + \beta^l = \gamma^l(\mathbf{W}_p^l * \mathbf{X}_c^l) + \beta^l, \qquad (2)$$

$$\gamma^l = \text{Avg}(\mathbf{W}_p^l * \mathbf{P}_\gamma^l), \qquad \beta^l = \text{Avg}(\mathbf{W}_p^l * \mathbf{P}_\beta^l), \qquad (3)$$

in which $*$ denotes the convolution operations, $l$ is the layer number, $\mathbf{X}_p$ is the input to the passport layer and $\mathbf{X}_c$ is the input to the convolution layer. $\mathbf{O}()$ is the corresponding linear transformation of outputs, while $\mathbf{P}_\gamma^l$ and $\mathbf{P}_\beta^l$ are the passports used to derive scale factor and bias term respectively. Fig. 2a delineates the architecture of digital passport layers used in a *ResNet* layer.

**Remark**: for DNN models trained with passport $\mathbf{s}_e = \{\mathbf{P}_\gamma^l, \mathbf{P}_\beta^l\}^l$, their *inference performances* $\mathcal{M}(\mathbb{N}[\mathbf{W}, \mathbf{s}_e], \mathbf{D}_t, \mathbf{s}_t)$ depend on the running time passports $\mathbf{s}_t$ i.e.

$$\mathcal{M}(\mathbb{N}[\mathbf{W}, \mathbf{s}_e], \mathbf{D}_t, \mathbf{s}_t) = \begin{cases} \mathcal{M}_{\mathbf{s}_e}, & \text{if } \mathbf{s}_t = \mathbf{s}_e, \\ \overline{\mathcal{M}_{\mathbf{s}_e}}, & \text{otherwise.} \end{cases} \qquad (4)$$

If the genuine passport is not presented $\mathbf{s}_t \neq \mathbf{s}_e$, the running time performance $\overline{\mathcal{M}}_{\mathbf{s}_e}$ is significantly deteriorated because the corresponding scale factor $\gamma$ and bias terms $\beta$ are calculated based on the wrong passports. For instance, as shown in Fig. 2b, a proposed DNN model presented with valid passports (green) will demonstrate almost identical accuracies as to the original DNN model (red). In contrast, the same proposed DNN model presented with counterfeit passports (blue), the accuracy will deteriorate to merely about 10% only.

**Remark**: the gist of the proposed passport layer is to enforce *dependence* between scale factor, bias terms and network weights. As shown by the Proposition 2, it is this dependence that validates the required non-invertibility to defeat ambiguity.

**Proposition 2** (*Non-invertible process*). A DNN ownership verification scheme $\mathcal{V}$ as in Definition 1 is *non-invertible*, if

    I) the fidelity process outcome $F(\mathbb{N}[\mathbf{W}, \mathbf{T}, \mathbf{s}], \mathbf{D}_t, \mathcal{M}_t, \epsilon_f)$ depends either on the presented signature $\mathbf{s}$ or trigger set $\mathbf{T}$,

    II) with forged passport $\mathbf{s}_t \neq \mathbf{s}_e$, the DNN inference performance $\mathcal{M}(\mathbb{N}[\mathbf{W}, \mathbf{s}_e], \mathbf{D}_t, \mathbf{s}_t)$ in (Eq. 4) will deteriorate such that the discrepancy is larger than a threshold i.e. $|\mathcal{M}_{\mathbf{s}_e} - \overline{\mathcal{M}}_{\mathbf{s}_e}| > \epsilon_f$.

### 3.2   Sign of scale factors as signature

During learning the DNN, to further protect the DNN models ownership from insider threat (e.g. a former staff who establish a new start-up business with all the resources stolen from originator), one can enforce the scale factor $\gamma$ to take either positive or negative signs (+/-) as designated, so that it will form a unique signature string (like fingerprint). This process is done by adding the following *sign loss* regularization term into the combined loss (Eq. 1):

$$R(\gamma, \mathbf{P}, \mathbf{B}) = \sum_{i=1}^{C} \max(\gamma_0 - \gamma_i b_i, 0) \tag{5}$$

in which $\mathbf{B} = \{b_1, \cdots, b_C\} \in \{-1, 1\}^C$ consists of the designated binary bits for $C$ convolution kernels, and $\gamma_0$ is a positive control parameter (0.1 by default unless stated otherwise) to encourage the scale factors have magnitudes greater than $\gamma_0$.

It must be highlighted that the inclusion of sign loss (Eq. 5) enforces the scale factors $\gamma$ to take either positive or negative values, and the signs enforced in this way remain rather persistent against various adversarial attacks. This feature explains the superior robustness of embedded passports against ambiguity attacks by reverse-engineering shown in Sect. 4.2.

### 3.3   Ownership verification with passports

Taking advantages of the proposed passport-based approach, we design three new ownership verification schemes $\mathcal{V}$ that are summarized next and refer readers to Sect. 4 for the experiment results.

#### $\mathcal{V}_1$: Passport is distributed with the trained DNN model

Hereby, the *learning* process aims to minimize the combined loss function (Eq. 1), in which $\lambda_t = 0$ since trigger set images are not used in this scheme and the sign loss (Eq. 5) is added as the regularization term. The trained DNN model together with the passport are then distributed to legitimate users, who perform network *inferences* with the given *passport* fed to the passport layers as shown in Fig. 2a. The network ownership is automatically verified by the distributed passports. As shown in Table 2 and Fig. 3, this ownership verification is robust to DNN model modifications. Also, as shown in Fig. 4, ambiguity attacks are not able to forge a set of passport and signature that can maintain the DNN inference performance.

The downside of this scheme is the requirement to use passports during inferencing, which leads to extra computational cost by about 10% (see Sect. 4.3). Also the distribution of passports to the end-users is intrusive and imposes additional responsibility of guarding the passports safely.

#### $\mathcal{V}_2$: Private passport is embedded but not distributed

Herein, the *learning* process aims to simultaneously achieve *two goals*, of which the first is to minimize the original task loss (e.g. classification accuracy discrepancy) when *no passport* layers included; and the second is to minimize the combined loss function (Eq. 1) with passports regularization included. Algorithm-wise, this *multi-task learning* is achieved by alternating between the minimization of these two goals. The successfully trained DNN model is then distributed to end-users, who may perform network inference *without the need of passports*. Note that this is possible since passport layers are not included in the distributed networks. The ownership verification will be carried out only upon requested by the law enforcement, by adding the passport layers to the network in question and detecting the embedded sign signatures with unyielding the original network inference performances.

Compared with scheme $\mathcal{V}_1$, this scheme is easy to use for end-users since no passport is needed and no extra computational cost is incurred. In the meantime, this ownership verification is robust to removal attacks as well as ambiguity attacks. The downside, however, is the requirement to access the DNN weights and to append the passport layers for ownership verification, i.e. the disadvantages of white-box protection mode as discussed in [2]. Therefore, we propose to combine it with trigger-set based verification that will be described next.

#### $\mathcal{V}_3$: Both the private passport and trigger set are embedded but not distributed

This scheme only differs from scheme $\mathcal{V}_2$ in that, a set of trigger images is embedded in addition to the embedded passports. The advantage of this, as discussed in [2] is to probe and claim ownership

| | CIFAR10 | | | | CIFAR100 | | |
|---|---|---|---|---|---|---|---|
| **AlexNet$_\mathbf{p}$** | CIFAR10 | CIFAR100 | Caltech-101 | **AlexNet$_\mathbf{p}$** | CIFAR100 | CIFAR10 | Caltech-101 |
| Baseline (BN) | - (91.12) | - (65.53) | - (76.33) | Baseline (BN) | - (68.26) | - (89.46) | - (79.66) |
| Scheme $\mathcal{V}_1$ | 100 (90.91) | 100 (64.64) | 100 (73.03) | Scheme $\mathcal{V}_1$ | 100 (68.31) | 100 (89.07) | 100 (78.83) |
| Baseline (GN) | - (90.88) | - (62.17) | - (73.28) | Baseline (GN) | - (65.09) | - (88.30) | - (78.08) |
| Scheme $\mathcal{V}_2$ | 100 (89.44) | 99.91 (59.31) | 100 (70.87) | Scheme $\mathcal{V}_2$ | 100 (64.09) | 100 (87.47) | 100 (76.31) |
| Scheme $\mathcal{V}_3$ | 100 (89.15) | 99.96 (59.41) | 100 (71.37) | Scheme $\mathcal{V}_3$ | 100 (63.67) | 100 (87.46) | 100 (75.89) |
| **ResNet$_\mathbf{p}$-18** | | | | **ResNet$_\mathbf{p}$-18** | | | |
| Baseline (BN) | - (94.85) | - (72.62) | - (78.98) | Baseline (BN) | - (76.25) | - (93.22) | - (82.88) |
| Scheme $\mathcal{V}_1$ | 100 (94.62) | 100 (69.63) | 100 (72.13) | Scheme $\mathcal{V}_1$ | 100 (75.52) | 100 (95.28) | 99.99 (79.27) |
| Baseline (GN) | - (93.65) | - (69.40) | - (75.08) | Baseline (GN) | - (72.06) | - (91.83) | - (79.15) |
| Scheme $\mathcal{V}_2$ | 100 (93.41) | 100 (63.84) | 100 (71.07) | Scheme $\mathcal{V}_2$ | 100 (72.15) | 100 (90.94) | 100 (77.34) |
| Scheme $\mathcal{V}_3$ | 100 (93.26) | 99.98 (63.61) | 99.99 (72.00) | Scheme $\mathcal{V}_3$ | 100 (72.10) | 100 (91.30) | 100 (77.46) |

Table 2: Removal Attack (Fine-tuning): Detection/Classification accuracy (in %) of different passport networks where BN = batch normalisation and GN = group normalisation. (Left: trained with CIFAR10 and fine-tune for CIFAR100/Caltech-101. Right: trained with CIFAR100 and fine-tune for CIFAR10/Caltech-101.) Accuracy outside bracket is the signature detection rate, while in-bracket is the classification rate.

of the suspect DNN model through remote calls of service APIs. This capability allows one, first to claim the ownership in a black-box mode, followed by reassertion of ownership with passport verification in a white box mode. Algorithm-wise, the embedding of trigger set images is jointly achieved in the same minimization process that embeds passports in scheme $\mathcal{V}_2$. Finally, it must be noted that the embedding of passports in both $\mathcal{V}_2$ and $\mathcal{V}_3$ schemes are implemented through *multi-task learning tasks* where we adopted group normalisation [12] instead of batch normalisation [13] that is not applicable due to its dependency on running average of batch-wise training samples.

# 4 Experiment results

This section illustrates the experiment results of passport-based DNN models whereas the inference performances of various schemes are compared in terms of *robustness* to both removal attacks and ambiguity attacks. The network architectures we investigated include the well-known AlexNet and ResNet-18, which are tested with typical CIFAR10 and CIFAR100 classification tasks. These medium-sized public datasets allow us to perform extensive tests of the DNN model performances. Unless stated otherwise, all experiments are repeated 5 times and tested against 50 fake passports to get the mean inference performance. Also, to avoid confusion to the original AlexNet and ResNet models, we denote AlexNet$_\mathbf{p}$ and ResNet$_\mathbf{p}$-18 as our proposed passport-based DNN models.

## 4.1 Robustness against removal attacks

### Fine-tuning

Table 2 shows that the signatures are detected at near to 100% accuracy for all the ownership verification schemes in the original task. Even after fine-tuning the proposed DNN models for a new task (e.g. from CIFAR10 to Caltech-101), almost 100% accuracy are still maintained. Note that a detected signature is claimed only *iff* all the binary bits are exactly matched. We ascribe this superior robustness to the unique controlling nature of the scale factors — in case that a scale factor value is reduced near to zero, the channel output will be virtually zero, thus, its gradient will vanish and lose momentum to move towards to the opposite value. Empirically we have not observed counter-examples against this explanation[2].

### Model pruning

The aim of model pruning is to reduce redundant parameters without compromise the performance. Here, we adopt the *class-blind* pruning scheme in [14], and test our proposed DNN models with different pruning rates. Figure 3 shows that, in general, our proposed DNN models still maintained near to 100% accuracy even 60% parameters are pruned, while the accuracy of testing data drops around 5%-25%. Even if we prune 90% parameters, the accuracy of our proposed DNN models are still much higher than the accuracy of testing data. As said, we ascribe the robustness against model pruning to the superior persistence of signatures embedded in the scale factor signs (see Sect. 3.2).

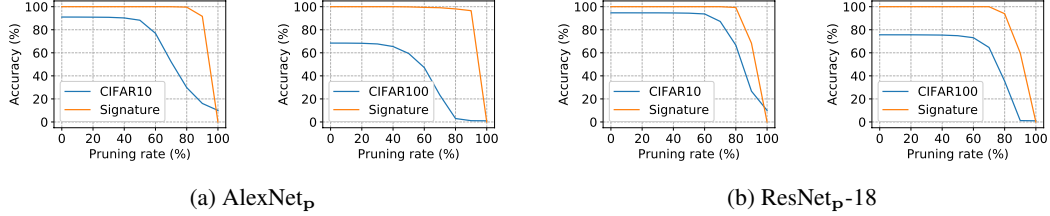

(a) AlexNet$_\mathbf{p}$          (b) ResNet$_\mathbf{p}$-18

Figure 3: Removal Attack (Model Pruning): Classification accuracy of our passport-based DNN models on both CIFAR10/CIFAR100 and signature detection accuracy against different pruning rates.

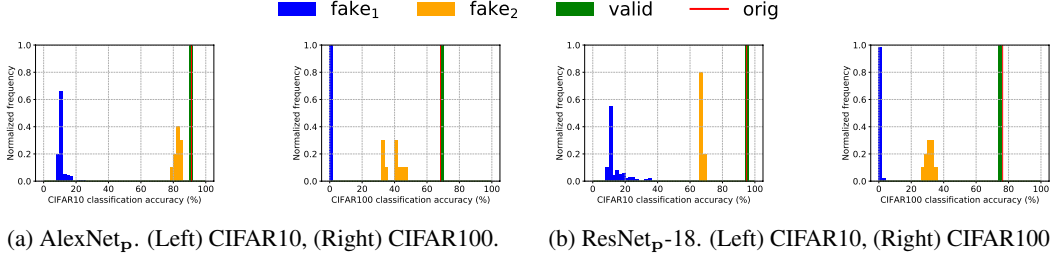

(a) AlexNet$_\mathbf{p}$. (Left) CIFAR10, (Right) CIFAR100.     (b) ResNet$_\mathbf{p}$-18. (Left) CIFAR10, (Right) CIFAR100.

Figure 4: Ambiguity Attack: Classification accuracy of our passport networks with valid passport, *random attack* ($fake_1$) and *reversed-engineering attack* ($fake_2$) on CIFAR10 and CIFAR100.

| Ambiguity attack modes | Attackers have access to | Ambiguous passport construction methods | Invertibility (see Def. 1.V) | Verification scheme $\mathcal{V}_1$ | Verification scheme $\mathcal{V}_2$ | Verification scheme $\mathcal{V}_3$ |
|---|---|---|---|---|---|---|
| $fake_1$ | $W$ | - Random passport $P_r$ | - $F(P_r)$ fail, by large margin | Large accuracy ↓ | Large accuracy ↓ | Large accuracy ↓ |
| $fake_2$ | $W, \{D_r; D_t\}$ | - Reverse engineer passport $P_e$ | - $F(P_e)$ fail, by moderate margin | Moderate accuracy ↓ | Moderate accuracy ↓ | Moderate accuracy ↓ |
| $fake_3$ | $W, \{D_r; D_t\}, \{P, S\}$ | - Reverse engineer passport $\{P_e; S_e\}$ by exploiting original passport $P$ & sign string $S$ | - if $S_e = S$: $F(P_e)$ pass, with negligible margin<br>- if $S_e \neq S$: $F(P_e)$ fail, by moderate to huge margin | refer to Fig. 5 | refer to Fig. 5 | refer to Fig. 5 |

Table 3: Summary of overall passport network performances in Scheme $\mathcal{V}_1$, $\mathcal{V}_2$ and $\mathcal{V}_3$, respectively under three different ambiguity attack modes, $fake$.

## 4.2 Resilience against ambiguity attacks

As shown in Fig. 4, the accuracy of our proposed DNN models trained on CIFAR10/100 classification task is significantly depending on the presence of either valid or counterfeit passports — the proposed DNN models presented with valid passports demonstrated almost identical accuracies as to the original DNN model. Contrary, the same proposed DNN model presented with invalid passports (in this case of $fake_1$ = random attack) achieved only 10% accuracy which is merely equivalent to a random guessing. In the case of $fake_2$, we assume that the adversaries have access to the original training dataset, and attempt to reverse-engineer the scale factor and bias term by freezing the trained DNN weights. It is shown that in Fig. 4, reverse-engineering attacks are only able to achieve, for CIFAR10, at best 84% accuracy on AlexNet$_\mathbf{p}$ and 70% accuracy on ResNet$_\mathbf{p}$-18. While in CIFAR100, for $fake_1$ case, attack on both our proposed DNN models achieved only 1% accuracy; for $fake_2$ case, this attack only able to achieve 44% accuracy for AlexNet$_\mathbf{p}$ and 35% accuracy for ResNet$_\mathbf{p}$-18.

Table 3 summarizes the accuracy of the proposed methods under three ambiguity attack modes, $fake$ depending on attackers' knowledge of the protection mechanism. It shows that all the corresponding passport-based DNN models accuracies are deteriorated to various extents. The ambiguous attacks are therefore defeated according to the fidelity evaluation process, $F()$. We'd like to highlight that even under the most adversary condition, i.e. freezing weights, maximizing the distance from the original passport $P$, and minimizing the accuracy loss (in layman terms, it means both the original passports and scale signs are exploited due to insider threat, and we class this as $fake_3$), attackers are still unable to use new (modified) scale signs without compromising the network accuracies. As shown in Fig. 5, with 10% and 50% of the original scale signs are modified, the CIFAR100 classification accuracy drops about 5% and 50%, respectively. In case that the original scale sign remains unchanged, the DNN model ownership can be easily verified by the pre-defined string of signs. Also, Table 3 shows that attackers are unable to exploit $D_t$ to forge ambiguous passports.

Based on these empirical studies, we decide to set the threshold $\epsilon_f$ in Definition 1 as 3% for AlexNet$_\mathbf{p}$ and 20% for ResNet$_\mathbf{p}$-18, respectively. By this fidelity evaluation process, any potential ambiguity

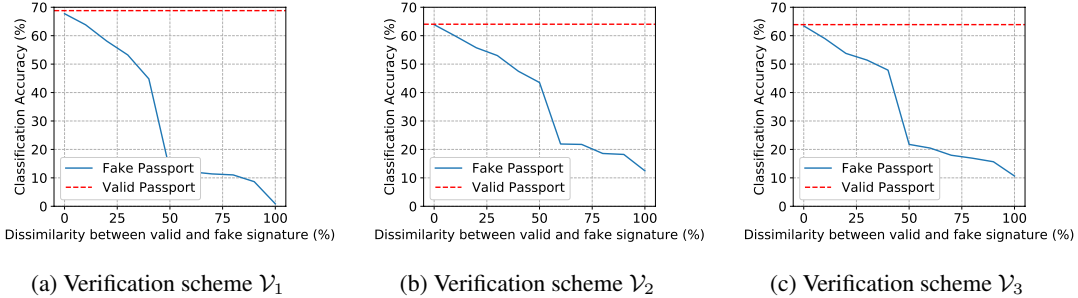

|  | (a) Verification scheme $\mathcal{V}_1$ | (b) Verification scheme $\mathcal{V}_2$ | (c) Verification scheme $\mathcal{V}_3$ |

Figure 5: Ambiguity Attack: Classification accuracy on CIFAR100 under insider threat ($fake_3$) on three verification schemes. It is shown that when a correct signature is used, the classification accuracy is intact, while for a partial correct signature (sign scales are modified around 10%), the performance will immediately drop around 5%, and a totally wrong signature will obtain a meaningless accuracy (1%-10%). Based on the threshold $\leq \epsilon_f = 3\%$ for AlexNet$_\mathbf{p}$ and by the fidelity evaluation process $F$, any potential ambiguity attacks (even with partially correct signature) are effectively defeated.

|  | Scheme $\mathcal{V}_1$ | Scheme $\mathcal{V}_2$ | Scheme $\mathcal{V}_3$ |
|---|---|---|---|
| Training | - Passport layers added<br>- Passports needed<br>- 15%-30% more training time | - Passport layers added<br>- Passports needed<br>- 100%-125% more training time | - Passport layers added<br>- Passports & Trigger set needed<br>- 100%-150% more training time |
| Inferencing | - Passport layers & Passports needed<br>- 10% more inferencing time | - Passport layers & Passport NOT needed<br>- NO extra time incurred | - Passport layers & Passport NOT needed<br>- NO extra time incurred |
| Verification | - NO separate verification needed | - Passport layers & Passports needed | - Trigger set needed (black-box verification)<br>- Passport layers & Passports needed (white-box verification) |

Table 4: Summary of our proposed passport networks complexity for $\mathcal{V}_1$, $\mathcal{V}_2$ and $\mathcal{V}_3$ schemes.

attacks are effectively defeated. In summary, extensive empirical studies have shown that it is impossible for adversaries to maintain the original DNN model accuracies by using counterfeit passports, regardless of they are either randomly generated or reverse-engineered with the use of original training datasets. This passport dependent performances play an indispensable role in designing secure ownership verification schemes that are illustrated in Sect. 3.3.

### 4.3 Network Complexity

Table 4 summarizes the complexity of passport networks in various schemes. We believe that it is the computational cost at the inference stage that is required to be minimized, since network inference is going to be performed frequently by the end users. While extra costs at the training and verification stages, on the other hand, are not prohibitive since they are performed by the network owners, with the motivation to protect the DNN model ownerships. Nonetheless, we tested a larger network (i.e. ResNet$_\mathbf{p}$-50) and its training time increases 10%, 182% and 191% respectively for $\mathcal{V}_1$, $\mathcal{V}_2$ and $\mathcal{V}_3$ schemes. This increase is consistent with those smaller models i.e. AlexNet$_\mathbf{p}$ and ResNet$_\mathbf{p}$-18.

## 5 Discussions and conclusions

Considering billions of dollars have been invested by giant and start-up companies to explore new DNN models virtually every second, we believe it is imperative to protect these inventions from being stolen. While ownership of DNN models might be resolved by registering the models with a centralized authority, it has been recognized that these regulations are inadequate and technical solutions are urgently needed to support the law enforcement and juridical protections. It is this motivation that highlights the unique contribution of the proposed method in unambiguous verification of DNN models ownerships.

Methodology-wise, our empirical studies re-asserted that over-parameterized DNN models can successfully learn multiple tasks with arbitrarily assigned labels and/or constraints. While this assertion has been theoretically proved [15] and empirically investigated from the perspective of network generalization [16], its implications to network security in general remain to be explored. We believe the proposed modulation of DNN performance based on the presented passports will play an indispensable role in bringing DNN behaviours under control against adversarial attacks, as it has been demonstrated for DNN ownership verifications.

## Acknowledgement

This research is partly supported by the Fundamental Research Grant Scheme (FRGS) MoHE Grant FP021-2018A, from the Ministry of Education Malaysia. Also, we gratefully acknowledge the support of NVIDIA Corporation with the donation of the Titan V GPU used for this research.

## Footnotes

[1]Learning hyper-parameters such as learning rate and the type of optimization methods are considered irrelevant to ownership verifications, and thus they are not included in the formulation.

[2]A rigorous proof of this argument is under investigation and will be reported elsewhere.

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
