[Supplementary Material · supp_mat2632.pdf]

**Appendix: Supplementary material -**
 **Rethinking Deep Neural Network Ownership Verification:**
 **Embedding Passports to Defeat Ambiguity Attacks**

## A  Invertibility of DNN ownership verification schemes

**Proposition 1** (*Invertible process*). For a DNN ownership verification scheme $\mathcal{V}$ as in Definition 1, if the fidelity process $F()$ is independent of either the signature $\mathbf{s}$ or trigger set $\mathbf{T}$, then there always exists an invertible process $I()$ i.e. the scheme is invertible $\mathcal{V}^I = (E, F, V, I \neq \emptyset))$.

*Proof.* for a trained network $\mathbb{N}[\hat{\mathbf{W}}, \mathbf{T}, \mathbf{s}]$ with signature $\mathbf{s}$ and/or trigger set $\mathbf{T}$ embedded, the invert process $I()$ can be constructed with the following steps:

1. maintain the optimal weights $\hat{\mathbf{W}}$ unchanged;

2. minimize the detection error (see III in Definition (1) in the main paper):

    i) forge the *feature-based* watermarks $\mathbf{s}' = \{\mathbf{P}', \mathbf{B}'\}$ by minimizing the distance $\{\mathbf{P}', \mathbf{B}'\} = arg \min_{\mathbf{P}, \mathbf{B}} D_f(f_e(\hat{\mathbf{W}}, \mathbf{P}), \mathbf{B})$. *Remark*: attackers have to take $\mathbf{B}' \neq \mathbf{B}$, and in case that the watermark signature $\mathbf{B}$ is unknown, attackers may assign random signature $\mathbf{B}'$, whose the probability of collision $\mathbf{B}' = \mathbf{B}$ is then exponentially low.

    ii) forge the trigger set $T' = \{\mathbf{X}'_T, \mathbf{y}'_T\}$ by minimizing the (cross-entropy) loss $\{\mathbf{X}'_T, \mathbf{y}'_T\} = arg \min_{\mathbf{X}_T, \mathbf{y}_T} L_c\big(f(\hat{\mathbf{W}}, \mathbf{X}_T), \mathbf{y}_T\big)$ between the prediction and the target labels.

3. fidelity evaluation is fulfilled since it is independent to both the forged signatures and trigger set, thus remain unchanged.

*Remark*: during the minimization of detection error, there is *no need of training data* which is not used in step 2 at all;

*Remark*: during the minimization of detection error, the *computational cost is minor* since the dimensionality of the optimization parameters i.e. $\{\mathbf{P}', \mathbf{B}'\}$ or $\mathbf{X}'_T, \mathbf{y}'_T$ is order of magnitude smaller, as compared to the number of DNN weights $\hat{\mathbf{W}}$. $\qquad\square$

**Proposition 2** (*Non-invertible process*). A DNN ownership verification scheme $\mathcal{V}$ as in Definition 1 is *non-invertible*, if

I) the fidelity process outcome $F\big(\mathbb{N}[\mathbf{W}, \mathbf{T}, \mathbf{s}], \mathbf{D}_t, \mathcal{M}_t, \epsilon_f\big)$ depends either on the presented signature $\mathbf{s}$ or trigger set $\mathbf{T}$,

II) with forged passport $\mathbf{s}_t \neq \mathbf{s}_e$, the DNN performance $\mathcal{M}(\mathbb{N}[\mathbf{W}, \mathbf{s}_e], \mathbf{D}_t, \mathbf{s}_t)$ in (4) is deteriorated such that the discrepancy is larger than a threshold i.e. $|\mathcal{M}_{\mathbf{s}_e} - \overline{\mathcal{M}}_{\mathbf{s}_e}| > \epsilon_f$.

*Proof.* Since using forged passports the DNN model performance is significantly deteriorated such that $|\mathcal{M}_{\mathbf{P}_e} - \overline{\mathcal{M}}_{\mathbf{P}_e}| > \epsilon_f$, it immediately follows, from the definition of invertible verification schemes, that the scheme in question is non-invertible. $\qquad\square$

## B  Ambiguity attacks on conventional DNN watermarking methods

This section illustrates experiment results related to the ambiguity attacks on existing watermarking methods, in particular, the feature-based [1] and the trigger-set based method [2]. We thank both authors for making the source codes publicly available at https://github.com/yu4u/dnn-watermark [1], and https://github.com/adiyoss/WatermarkNN [2].

### B.1  Ambiguity attacks on feature-based method [1]

In this experiment, we first train a DNN model embedded with watermarks as described in [1], then we conduct the ambiguity attacks as follows.

The loss function adopted in [1] uses the following binary cross entropy for the embedding regularizer:

$$E_R(W) = -\sum_{j=1}^{T}(b_j \log(y_j) + (1 - b_j)\log(1 - y_j)),\qquad(6)$$

in which $y_j = \sigma(\sum_i X_{ji}w_i)$ is the extracted feature with $\sigma(\cdot)$ is the sigmoid function. In order to forge watermark $X$ for a given signature $b_j$, we first freeze the weights $w_i$ of the watermarked DNN model, and minimize the loss (Eq. 6) with respect to the new binary signatures $b'_j$.

(a) Distribution of $X_{ji}$.  (b) Histogram of $\sum_i X_{ji}w_i$.  (c) $X_{ji}$ with fine-tuning.

Figure 6: Distributions of $X_{ji}$ and extracted features.

Figure 6a illustrates the distributions of fake watermarks $X_{ji}$ together with the real watermarks, which are hardly distinguishable from each other. In terms of the extracted features $\sum_i X_{ji}w_i$, their distributions are different from the original watermarks, but it is impossible to tell the difference after thresholding for the purpose of ownership verification. Also, Figure 6c illustrates that the distribution of $X_{ji}$ is not much affected by the fine-tuning process which aims to modify the DNN weights for transfer learning purposes (see Table 3).

Following [1], we detect watermarks by comparing the extracted binary strings w.r.t. the designated one by measuring the successful detection rate. As summarized in Table 3, all the fake watermarks of size (256-bit) are successfully detected. We also fine-tune the DNN model by adjusting the network weights at all layers for new classification tasks (i.e. Caltech-101 and CIFAR100), where fake watermarks are still detectable with 100% detection rate demonstrating robustness against fine-tuning too.

|  | Detection rate (feature-based) | |
|---|---|---|
| Transfer Learning Task | Real watermark | Fake watermark |
| CIFAR10 (90.97%) –> CIFAR100 (64.25%) | 100% - 100% | 100% - 100% |
| CIFAR10 (90.97%) –> Caltech-101 (74.08%) | 100% - 100% | 100% - 100% |

Table 3: Detection rate of embedded watermarks, before and after fine-tuning. In-bracket values indicate average test accuracies of the original task and the transfer-learning task as in [1]

.

Note that since $w_i$ are fixed, we do not need to include the original (cross-entropy) loss measured with the training images, which is a constant during the optimization. This simplicity allows the forging of $X_{ji}$ converge very rapidly. The overall optimization took about 50 iterations in 50 seconds, which merely constitutes a minor fraction (2.5%) of the training time for the original task.

## B.2 Ambiguity attacks on trigger-set based method [2]

In this experiment, we first follow [2] to train the DNN model with trigger set images embedded as watermarks, and then we conduct the ambiguity attacks as follows.

In order to construct the adversarial trigger set images by minimizing the cross-entropy loss between the predicted labels and the target labels, we adopt a simple approach which adds trainable noise components to randomly selected base images using the following steps:

1. Randomly select a set of $N$ base images $T_b$ as shown in Figure 7a;
2. Make random noisy patterns of the same size $T_n$ as trainable parameters;
3. Use the summed components $\mathbf{X}_T = T_b + \eta T_n$ as the trigger set images, in which $\eta = 0.04$ to make the noise component invisible;
4. Randomly assign trigger set labels $\mathbf{y}_T$;
5. Minimize the cross-entropy cost $L_c\big(f(\hat{\mathbf{W}}, \mathbf{X}_T), \mathbf{y}_T\big)$ w.r.t. the trainable parameter $T_n$.

   *Remark*: DNN parameters $\hat{\mathbf{W}}$ are fixed during the optimization, and thus, the original training data is not needed.

(a) base image $T_b$

(b) noise component $T_n$

(c) optimized $X_T$

Figure 7: Sample of the new trigger set images

Figure 8: Distribution of the real $T_b$ and fake $X_T$ trigger set

Figure 7c illustrates the final optimized $\mathbf{X}_T$. where all of them are correctly classified as the assigned labels i.e. $\mathbf{y}_T$. Visually, these forged trigger set images (Figure 7c) are hardly distinguishable from the original ones (Figure 7a). In terms of histogram distributions, they are indistinguishable too (Figure

|  | Detection rate (trigger-set) | |
| Transfer Learning Task | Real watermark | Fake watermark |
| --- | --- | --- |
| CIFAR10 (91.03%) –> CIFAR100 (65.20%) | 100% - 25.0% | 100% - 27.8% |
| CIFAR10 (91.03%) –> Caltech-101 (75.06%) | 100% - 43.6 % | 100% - 46.8% |

Table 4: Detection rate of watermarks, before and after fine-tuning. In-bracket values indicate average test accuracy of the corresponding tasks from the DNN model that adopted watermark technique from [2]

8). As shown in Table 4, both the trigger set and forged images are 100% correctly labeled with assigned adversarial labels. This indistinguishable situation casts doubt on ownership verification by trigger set images alone.

After fine-tuned to other classification tasks, however, the classification accuracies of both trigger set and forged images deteriorated drastically yet the detection rate of forged images is slightly better than that of the original trigger set images. We ascribed this improvement to the ambiguity attack procedures outlined above.

In terms of the computational cost, the overall optimization takes about 100 epochs of fake trigger set in 100 seconds, which merely constitutes a minor fraction (5%) of the training time for the original task.

## C  Embedding passports into DNN models

This section illustrates the empirical study of passport-embedded networks, with focuses on *convergence* and *effectiveness* of passport layers as well as its *robustness* against both modifications (i.e. fine-tuning & pruning) and ambiguity attacks. Please see Section 3.1 in the main paper for description.

### C.1  General performances

**Modulated performance**: the gist of embedding digital passports is to design and train DNN models in a way such that, the network performances of the original task will be significantly *deteriorated due to forged signatures*. As shown in Figure 9, for both AlexNet and ResNet trained for CIFAR10 & CIFAR100 classification tasks, there are significant performance margins ranging from 3% up to 80% depending on the presence of either valid or fake passports — DNN model presented with valid passports demonstrated almost identical accuracies as that of the original network (over 90%), while the same DNN model presented with fake passports achieved about 10% classification rates which is merely equivalent to a random guessing.

Regardless whether the fake passports are either randomly picked up ($fake_1$, Appendix C.3.3) or reverse-engineered ($fake_2$, Appendix C.3.4), the modulated DNN model performances play indispensable roles in distinguishing valid passports from the forged ones and defeating potential ambiguity attacks. For instance, one may set the threshold $\epsilon_f$ in Definition 1 of the main paper as 3% and 20%, respectively, for AlexNet and ResNet respectively. By this fidelity evaluation process, any potential ambiguity attacks are then effectively defeated.

**Convergence**: the introduction of the proposed passport layers (Figure 2a in the main paper) does not hinder the convergence of DNN learning process. As shown in Figure 10, we observe that the test accuracies converge in synchronization with the network weights, and computed linear transformation parameters $\gamma$ and $\beta$ which all stagnate in the later learning phase when the learning rate is reduced from 0.01 to 0.0001 (see Appendix F for hyper-parameter settings).

**Public and hidden DNN parameters**[3]

By introducing the passport layers, we essentially separate the DNN parameters into two types: the *public* convolution layer parameters $\mathbf{W}$ and the *hidden* passport layer scale factor $\gamma$ and bias terms $\beta$ (see Eq. (4) in the main paper). The learning of each of these parameter types are different too. On one hand, the distribution of the convolution layer weights seems identical to that of the original DNN without passport layers (Figure 11a). However, we must emphasize that information about the passports are embedded into weights $\mathbf{W}$ in the sense that following constraints are enforced once the learning is done:

$$\text{Avg}(\mathbf{W}_p^l * \mathbf{P}_\gamma^l) = c_\gamma^l, \qquad \text{Avg}(\mathbf{W}_p^l * \mathbf{P}_\beta^l) = c_\beta^l, \tag{7}$$

where $c_\gamma^l, c_\beta^l$ are two constants of converged parameters $\gamma^l, \beta^l$.

On the other hand, the distribution of the hidden parameters are affected by the adoption of sign loss (Eq. 5). Clearly the scale factors are enforced to take either positive or negative values far from zero (Figure 11b). We also observe that the sign of scale factors remain rather persistent against various adversarial attacks (Appendix C.3). An additional benefit of enforcing non-zero magnitudes of scale factors is to ensure the non-zero channel outputs and slightly improve the performances. Correspondingly the distribution of bias terms becomes more balanced with the sign loss regularization (Eq. 5) included, whereas the original bias terms are mainly negative valued (Figure 11c).

### C.2  Training and inference time of a passport DNN model

Table 5 shows the training and inference time of each scheme on AlexNet and ResNet, respectively using one NVIDIA Titan V. In both of the DNN architectures, the inference time of the baseline, scheme $\mathcal{V}_2$, scheme $\mathcal{V}_3$ are almost the same as to the execution time because all of them didn't use passport to calculate $\gamma$ and $\beta$. However, scheme $\mathcal{V}_1$ is slightly slower (about 10%) compared to the baseline because of the extra computational cost of $\gamma$ and $\beta$ calculation from the passport. Training time of

(a) AlexNet. Left: CIFAR10, right: CIFAR100.

(b) ResNet. Left: CIFAR10, right: CIFAR100.

Figure 9: DNN model performances with valid passports and two different types of fake passports i.e. *random attack* is $fake_1$ (Appendix C.3.3) and *ambiguity attack* is $fake_2$ (Appendix C.3.4). Accuracies of each DNN architecture are averaged over 5 runs, where each trained model is tested against 50 $fake_1$ and 5 $fake_2$ passports. Accuracies with valid passports and original DNN (wo. passport) are too close to separate in histograms.

|  | CIFAR10 | |
|---|---|---|
|  | T | I |
| AlexNet Baseline | 8.445 | 0.834 |
| AlexNet $\mathcal{V}_1$ | 10.745 | 0.912 |
| AlexNet $\mathcal{V}_2$ | 19.010 | 0.830 |
| AlexNet $\mathcal{V}_3$ | 21.372 | 0.881 |

|  | CIFAR10 | |
|---|---|---|
|  | T | I |
| ResNet baseline | 31.09 | 1.71 |
| ResNet $\mathcal{V}_1$ | 36.67 | 1.94 |
| ResNet $\mathcal{V}_2$ | 67.21 | 1.87 |
| ResNet $\mathcal{V}_3$ | 70.69 | 1.88 |

Table 5: Training (T) and Inference (I) time of each scheme on AlexNet (left) and ResNet (right) using one *NVIDIA Titan V*. The values are in seconds/epoch.

scheme $\mathcal{V}_1$, scheme $\mathcal{V}_2$ and scheme $\mathcal{V}_3$ are slower than the baseline about 18%(ResNet)/27%(AlexNet), 116%(ResNet)/125% and 127%(ResNet)/153%, respectively. Scheme $\mathcal{V}_2$ and scheme $\mathcal{V}_3$ are slower about 2x than scheme $\mathcal{V}_1$ due to the multi task training scheme.

## C.3    More experiments of the proposed method against various attacks

### C.3.1. - Robustness against fine-tuning

In this experiment, we repeatedly trained each model five times with designated scale factor signs embedded into both AlexNet and ResNet networks. Passport signatures are then detected at 100% detection rates for all three ownership verification schemes introduced in Section 3.3 in the main paper

(a) Test accuracies          (b) Weights update

(c) Scale factors          (d) Bias terms

Figure 10: (a) Convergences of test accuracies, (b) weight updates, (c) scale factors, and (d) bias terms of first 10 channels in Conv4 of AlexNet. x-axis: training epochs; y-axis: see captions of subfigures.

(a) Weight distribution      (b) Scale factor distribution      (c) Bias term distribution

Figure 11: Comparison of the distributions of (a) network weights, (b) scale factors, and (c) bias terms between the original and passport DNN (Conv4 of AlexNet)

and Appendix E, Table 2 (in the main paper) shows that even after fine-tuning for other classification tasks (e.g. from CIFAR10 → Caltech-101), the 100% detection rates of embedded passports are still maintained.

Note that a detected passport signature is claimed only if all binary bits are exactly matched. We ascribe this superior robustness to the stringent controlling nature of scale factors — in the case that a scale factor value is reduced to near zero, the channel output will be virtually zero. Thus, its gradient will vanish and lose momentum to move towards to the opposite sign. Empirically there is no evidence observed otherwise[4].

| | Trigger Set Detection | To CIFAR10 | To CIFAR100 | To Caltech-101 |
|---|---|---|---|---|
| AlexNet CIFAR10 | 100% | - | 24.67% | 57.67% |
| AlexNet CIFAR100 | 100% | 36.00% | - | 78.67% |
| ResNet CIFAR10 | 100% | - | 12.50% | 13.67% |
| ResNet CIFAR100 | 100% | 6.33% | - | 4.67% |

Table 6: Detection rate of the trigger set images (before and after fine-tuning) used in scheme $\mathcal{V}_3$ to complement passport-based verifications.

Figure 12: AlexNet: It can be seen that the performance deteriorates with randomly flipped scale factor signs. Left to right: flip one layer, two layers and three layers, respectively. Top row is CIFAR10 and bottom row is CIFAR100 dataset.

Table 6 shows the trigger set image detection rate before and after fine-tuning. Note that passports are not used in this case, therefore, the detection rate of the trigger set labels deteriorated drastically after fine-tuning. Nevertheless, trigger set images can still be used in scheme $\mathcal{V}_3$ to complement the white-box passport-based verification approach.

**C.3.2. - Robustness against pruning**

In this experiment, we test the passport-embedded DNN models against a certain percentage of the trained DNN weights being pruned. This type of weight pruning strategy has been adopted for network compression, which manage to maintain the original network performances even though the pruning percentage are high. For instance, an AlexNet network trained with CIFAR10 can be pruned up to almost 60% of its weights without significantly scarifying its performance.

Figure 4 (in the main article) shows that, the embedding of passport signatures in the sign of scale factors are rather persistent against pruning of the network weights. For instance, for CIFAR10 classification, a passport signature detection accuracy near 100% is maintained at the pruning percentage around 60%, and detection rate remains at 70% accuracy even though 90% of the network weights are pruned.

**C.3.3. - Robustness against random attacks**

The following experiments aim to disclose the dependence of the original task performances with respect to the crucial parameter *scale factors*, and specifically, its positive/negative signs.

In the first experiment, for the passport-embedded DNN models, we simulate random attacks by flipping the signs of certain randomly selected scale factors and then measure the performance. It turns out that the final performance are sensitive to the change of signs — majority of the DNN model performances drop significantly as long as more than (at least) 50% of scale factors have flipped signs as shown in Figures 12 and 13, respectively. The deteriorated performances are more pronounced when the passports are embedded in either all the three convolution layers (3-4-5) in AlexNet (right-most column in Figure 12) or the last blocks in ResNet (Figure 13), whose performances drop to about 10% and 1% respectively.

Figure 13: ResNet: It can be seen that the performance deteriorates with randomly flipped scale factor signs. Left is CIFAR10 and right is CIFAR100 dataset.

(a) CIFAR10 (left: AlexNet; right: ResNet)

(b) CIFAR100 (left: AlexNet; right: ResNet)

Figure 14: Performance of (a) CIFAR10 and (b) CIFAR100 when adversaries try to forge a new signature by a certain % of dissimilarity with the original signature.

The simulation results summarized in Figures 12 and 13 are in accordance to the results illustrates in Figure 9, which shows that the performance of passport-embedded DNNs under the attack of randomly assigned passport signatures. The poor performances measured for both AlexNet and ResNet on CIFAR10/CIFAR100 tasks are in the range of [10%, 30%] and [1%, 3%] respectively.

## C.3.4. - Robustness against ambiguity attacks

In this experiment, we further assume the adversaries have the access to original training data and thus are able to maximize the original task performance by reverse-engineering scale factors (i.e. flipping the sign (+/-) of the scale factor). The trained AlexNet/ResNet are used for this experiment, and it turns out the best performance the adversary can achieve is no more than 84%/70% for CIFAR10 and 40%/38% for CIFAR100 classifications respectively (see Figure 14).

In summary, extensive empirical studies show that it is impossible for adversaries to maintain the original DNN model performances by using fake passports, regardless of the fake passports are either randomly generated or reverse-engineered with the use of original training datasets. This passport

528 dependent performances play an indispensable role in designing secure ownership verification
529 schemes that are illustrated in Section 3.3 in the main paper.

530 **C.3.5. - Sign of scale factors $\gamma$ to encode signature $s$**

| Learned Parameters | | Signature $s$ | |
| --- | --- | --- | --- |
| Scale factor $\gamma$ | sign (+/-) | ASCII code | Character |
| -0.1113 | -1 | | |
| 0.2344 | 1 | | |
| 0.2494 | 1 | | |
| 0.4885 | 1 | 116 | t |
| -0.1021 | -1 | | |
| 0.3889 | 1 | | |
| -0.1225 | -1 | | |
| -0.3401 | -1 | | |
| -0.1705 | -1 | | |
| 0.3338 | 1 | | |
| 0.1884 | 1 | | |
| -0.1215 | -1 | 104 | h |
| 0.1620 | 1 | | |
| -0.1754 | -1 | | |
| -0.2698 | -1 | | |
| -0.1958 | -1 | | |
| -0.1007 | -1 | | |
| 0.3923 | 1 | | |
| 0.4288 | 1 | | |
| -0.1125 | -1 | 105 | i |
| 0.4355 | 1 | | |
| -0.1524 | -1 | | |
| -0.1073 | -1 | | |
| 0.1922 | 1 | | |
| -0.1999 | -1 | | |
| 0.2710 | 1 | | |
| 0.1599 | 1 | | |
| 0.2496 | 1 | 115 | s |
| -0.1345 | -1 | | |
| -0.1907 | -1 | | |
| 0.2326 | 1 | | |
| 0.1967 | 1 | | |

| Learned Parameters | | Signature $s$ | |
| --- | --- | --- | --- |
| Scale factor $\gamma$ | sign (+/-) | ASCII code | Character |
| -0.1657 | -1 | | |
| 0.1665 | 1 | | |
| 0.4633 | 1 | | |
| -0.2668 | -1 | 105 | i |
| 0.3830 | 1 | | |
| -0.1789 | -1 | | |
| -0.1077 | -1 | | |
| 0.1585 | 1 | | |
| -0.2257 | -1 | | |
| 0.2916 | 1 | | |
| 0.2169 | 1 | | |
| 0.1862 | 1 | 115 | s |
| -0.1146 | -1 | | |
| -0.1512 | -1 | | |
| 0.2288 | 1 | | |
| 0.3064 | 1 | | |

Table 7: Sample of the learned scale factor $\gamma$ and respective signs (+/-) from the 48 out of 256 channels from Conv5 AlexNet when we embed signature s = {this} and {is}.

531 In this section, we show how the sign (+/-) of scale factor $\gamma$ can be used to encode a signature $s$ such
532 as ASCII code. Table 7 shows an example of the learned scale factors and its respective sign when
533 we embed a signature $s = \{this\ is\ an\ example\ signature\}$ into the Conv5 of AlexNet by using
534 sign loss (Eq. 5). Note that the maximum size of an embedded signature is depending on the number
535 of the channels in a DNN model. For instance, in this paper, the Conv5 of AlexNet as shown in Table
536 10 has 256 channels, so the maximum signature capacity can be embedded is 256bits.

537 For ownership verification, the embedded signature $s$ can be revealed by decoding the learned sign of
538 scale factors. For example, in Table 7, every 8bits of the scale factor sign is decoded into ASCII code
539 as follow:

540    1. $\{-1,1,1,1,-1,1,-1,-1\} \rightarrow 116 \rightarrow t$

541    2. $\{-1,1,1,-1,1,-1,-1,-1\} \rightarrow 104 \rightarrow h$

542    3. $\{-1,1,1,-1,1,-1,-1,-1\} \rightarrow 105 \rightarrow i$

543    4. $\{-1,1,1,1,1,-1,-1,1,1\} \rightarrow 115 \rightarrow s$

544

545    4. $\{-1,1,1,-1,1,-1,-1,1\} \rightarrow 105 \rightarrow i$

546    3. $\{-1,1,1,1,-1,-1,1,1\} \rightarrow 115 \rightarrow s$

547 Note that, in this proposed method, similar character (e.g. {i} and {s}) appears in different position
548 of a string will have different scale factors.

|  | Signature $s$ | Accuracy (%) |
|---|---|---|
| AlexNet (baseline) | - | 91.12 |
| AlexNet | *this is an example signature* | 90.89 |
|  | *thhs iB an xxxpxX\| sigjature* | 82.83 |
|  | *qpCA2J$^O$Ec$\Delta$®o, * 1ay* | 11.44 |

Table 8: A comparison of the accuracy of AlexNet in CIFAR10 classification task when a correct (top), partially correct (middle) or totally wrong (bottom) signature is used.

Table 8 shows a comparison result when a correct signature, partial correct signature or total wrong signature is used in CIFAR10 classification task with AlexNet. It is shown that when a correct signature is used (i.e this is an example signature), the classification accuracy reached 90.89%, while for a partial correct signature, the performance is dropped to 82.23%, and a totally wrong signature will obtain a meaningless accuracy (11.44%). Based on the threshold $\epsilon_f = 3\%$ for AlexNet and by the fidelity evaluation process, any potential ambiguity attacks (even with partially correct signature) are effectively defeated.

 **D   Methods to generate passports**

<center>(a)                               (b)                               (c)</center>

Figure 15: Example of different types of passports: (a) random patterns, (b) fixed image and (c) random shuffled.

Figure 15 illustrates three different types of passports which have been investigated in our work:

   a) *random patterns*, whose elements are independently randomly generated according to the uniform distribution between [-1, 1].

   b) one selected image is fed through a trained DNN model with the same architecture, and the corresponding feature maps are collected. Then the selected *image* is used at the input layer and the *corresponding feature* maps are used at other layers as passports. We refer to passports generated as such the *fixed image* passport.

   c) a set of $N$ selected *images* are fed to a trained DNN model with the same architecture, and $N$ corresponding *feature maps* are collected at each layer. Among the $N$ options, only one is randomly selected as the passport at each layer. Specifically, for a set of $N$ images being applied to a DNN model with $L$ layers, there are altogether $N^L$ possible combinations of passports that can be generated. We refer to passports generated as such the *randomly shuffled* passports.

Since randomly shuffled passports allow strong protection and flexibility in the passport generation and distribution, we adopt this passport generation method for all the experiments reported in this paper. Specifically, 20 images are selected and fed to both *AlexNet* and *ResNet* that are used in our experiments. Passports at those corresponding convolution layers are then collected as possible passports. Some example of the features maps selected as the passports at different layers are illustrated in Figure 16.

Figure 16: *Randomly shuffled* passports in a 5-layered passport *AlexNet*. From left to right: Conv1 to Conv5 layers where the 4 passports in Conv2 to Conv5 corresponding to the first 4 channel of each layer.

 **E** **Ownership verification schemes** $\mathcal{V}_1, \mathcal{V}_2, and \mathcal{V}_3$

 **E.1 Overview**

 Taking advantages of the proposed passport embedding method, we design three ownership verifica-
 tion schemes which are summarized in Figure 3 of Section 3.3 in the main paper. Their respective
 merits and demerits, in terms of computational complexity, ease to use and protection strengths etc.
 are summarized in Table 9.

| | **Passport used** | Trigger set used for verification | Weights needed for verification | **Multi-task Learning** | $E$ | $M$ for $F$ | $V$ |
|---|---|---|---|---|---|---|---|
| $\mathcal{V}_1$ | Yes , inf. , / Yes , verif. | No | Yes | No | $\mathbb{N}[\mathbf{W}, \mathbf{s}_e]$ | $\mathcal{M}_{\mathbf{s}_e}$, if $\mathbf{s}_t = \mathbf{s}_e$, / $\overline{\mathcal{M}_{\mathbf{s}_e}}$, otherwise. | $V(\mathbb{N}[\mathbf{W}, \mathbf{s}_e])$ |
| $\mathcal{V}_2$ | No , inf. , / Yes , verif. | No | Yes | Yes | $\mathbb{N}[\mathbf{W}]$, inf. , / $\mathbb{N}[\mathbf{W}, \mathbf{s}_e]$, verif. | $\mathcal{M}_{\mathbf{s}_e}$, inf. , / $\mathcal{M}_{\mathbf{s}_e}$, if $\mathbf{s}_t = \mathbf{s}_e$, / $\overline{\mathcal{M}_{\mathbf{s}_e}}$, otherwise. | Not needed , inf. , / $V(\mathbb{N}[\mathbf{W}, \mathbf{s}_e])$, verif. |
| $\mathcal{V}_3$ | No , inf. , / Yes , verif. | Yes | No , verif.T , / Yes , verif.P | Yes | $\mathbb{N}[\mathbf{W}]$, inf. , / $\mathbb{N}[\mathbf{W}, \mathbf{T}, \mathbf{s}_e]$, verif. | $\mathcal{M}_{\mathbf{s}_e}$, inf. , / $\mathcal{M}_{\mathbf{s}_e}$, if $\mathbf{s}_t = \mathbf{s}_e$, / $\overline{\mathcal{M}_{\mathbf{s}_e}}$, otherwise. | Not needed, inf. , / $V(\mathbb{N}[\mathbf{W}, \mathbf{T}_e])$, verif.T / $V(\mathbb{N}[\mathbf{W}, \mathbf{s}_e])$, verif.P |

Table 9: A comparison of the features of the three passport-based ownership verification schemes depicted in Section 3.3 of the main paper. See Definition (1) for process $E, F, V$ and Eq. (4) in the main paper for the DNN model performances $M$. Notations: "inf." is network inference; "verif" is ownership verification; "verif.P" is verification by passport (white-box); "verif.T" is by trigger set samples (black-box).

 **E.2 Algorithms**

 Pseudo-code of the three verification schemes are illustrated in this section. For reproducibility of
 this work, we will make publicly available all source codes as well as the training / test datasets that
 are used in this paper, together with the camera-ready of the submission should the manuscript be
 accepted.

---

**Algorithm 1** Forward pass of a passport layer using scheme $\mathcal{V}_1$

---

1: **procedure** FORWARD $\mathcal{V}_1(X_c, W_p, P_\gamma, P_\beta)$
2:     $\gamma \leftarrow Avg(W_p * P_\gamma)$
3:     $\beta \leftarrow Avg(W_p * P_\beta)$
4:     $X_p \leftarrow W_p * X_c$
5:     $Y_p \leftarrow \gamma * O(X_p) + \beta$        ▷ O is a linear transformation such as BatchNorm
6:     **return** $Y_p$

---

**Algorithm 2** Forward pass of a passport layer using scheme $\mathcal{V}_2$ and $\mathcal{V}_3$

---

1: **procedure** FORWARD $\mathcal{V}_{23}(X_c, W_p, P_\gamma, P_\beta, \gamma_{publ}\ \beta_{publ}, idx)$
2:     **if** $idx = 0$ **then**
3:         $X_p \leftarrow W_p * X_c$
4:         $Y_p \leftarrow \gamma_{publ} * O(X_p) + \beta_{publ}$        ▷ $\gamma_{publ}$ and $\beta_{publ}$ is a public parameter
5:         **return** $Y_p$
6:     **else**
7:         **return** FORWARD $\mathcal{V}_1(X_c, W_p, P_\gamma, P_\beta)$

---

 Using Algorithm 4, we can extract a binarized version of *signature s* where positive $\gamma$ is 1 and
 negative $\gamma$ is 0 from model $M_p$. We can then decode $s$ into desired format such as ASCII code.
 Finally, we can claim ownership of the model $M_p$.

---
**Algorithm 3** Sign Loss
---
1: **procedure** SIGN LOSS($B^l, W_p^l, P_\gamma^l, \gamma_0$)
2:     $\gamma^l \leftarrow Avg(W_p^* P_\gamma^l)$
3:     $loss \leftarrow max(\gamma_0 - \gamma^l * B^l, 0)$       ▷ $\gamma_0$ is a positive constant, equals 0.1 as by default
4:     **return** loss
---

---
**Algorithm 4** Signature detection
---
1: **procedure** SIGNATURE DETECTION($W_p, P_\gamma$)
2:     $\gamma \leftarrow Avg(W_p * P_\gamma)$
3:     $signature \leftarrow sign(\gamma)$
4:     convert *signature* into binary
5:     decode binarized *signature* into desired format e.g. ascii
6:     match decoded *signature* with target signature
---

### E.3   Multi-task learning with private passports and/or trigger set images

The multi-task learning algorithms used for embedding passports in schemes $\mathcal{V}_2$ and $\mathcal{V}_3$ are summarized in Algorithm 6 which is similar to Algorithm 5.

It must be noted that the practical choice of formula (Eq. 2) is inspired by the well-known *Batch Normalization* (BN) layer which essentially applies the channel-wise linear transformation to the inputs[5]. Nevertheless BN is not applicable to multi-task learning tasks because of its dependency on running average of batch-wise training samples. When BN is used for multi-task learning, the test accuracy is significantly reduced even though the training accuracy seems optimized. We therefore adopted *group normalization* (GN) in the baseline DNN model for schemes $\mathcal{V}_2$ and $\mathcal{V}_3$ reported in Table 2[6].

---
**Algorithm 5** Training step for scheme $\mathcal{V}_1$
---
1: initialize a passport model $M_s$ with desired number of passport layers, $N_{pass}$
2: initialize passport keys $P$ in $M_s$
3: encode desired *signature* $s$ into binary to be embedded into signs of $\gamma_p$ of all passport layers
4: **for** number of training iterations **do**
5:     sample minibatch of $m$ samples $X$ $\{X^{(1)}, \cdots, X^{(m)}\}$ and targets $Y$ $\{Y^{(1)}, \cdots, Y^{(m)}\}$
6:     **if** enable backdoor **then**
7:         sample $t$ samples of $T$ and backdoor targets $Y_T$       ▷ $t = 2$, default by [2]
8:         concatenate $X$ with $T$, $Y$ with $Y_T$
9:     compute cross-entropy loss $L_c$ using $X$ and $Y$
10:     **for** $l$ in $N_{pass}$ **do**
11:         compute sign loss $R^l$ using $s^l$ and $\gamma_p^l$
12:     $R \leftarrow \sum_l^{N_{pass}} R^l$
13:     compute combined loss $L$ using $L_c$ and $R$
14:     backpropagate using $L$ and update $M_p$
---

**Algorithm 6** Training step for scheme $\mathcal{V}_2$ and $\mathcal{V}_3$

---

1: initialize a passport model $M_s$ with desired number of passport layers, $N_{pass}$
2: **if** enable trigger set **then**                                                                                      ▷ for scheme $\mathcal{V}_3$
3:     initialize trigger sets $T$
4:     initialize passport keys $P$ in $M_s$ using $T$
5: **else**
6:     initialize passport keys $P$ in $M_s$
7: encode desired *signature s* into binary to be embedded into signs of $\gamma_p$ of all passport layers
8: **for** number of training iterations **do**
9:     sample minibatch of $m$ samples $X \{X^{(1)}, ..., X^{(m)}\}$ and targets $Y \{Y^{(1)}, ..., Y^{(m)}\}$
10:     **if** enable backdoor **then**
11:         sample $t$ samples of $T$ and backdoor targets $Y_T$                              ▷ $t = 2$, default by [2]
12:         concatenate $X$ with $T$, $Y$ with $Y_T$
13:     **for** $idx$ in 0 1 **do**
14:         **if** $idx = 0$ **then**
15:             compute cross-entropy loss $L_c$ using $X$, $Y$ and $\gamma_{publ}$
16:         **else**
17:             compute cross-entropy loss $L_c$ using $X$ and $Y$
18:             **for** $l$ in $N_{pass}$ **do**
19:                 compute sign loss $R^l$ using $s^l$ and $\gamma_p^l$
20:     $R \leftarrow \sum_l^{N_{pass}} R^l$
21:     compute combined loss $L$ using $L_c$ and $R$
22:     backpropagate using $L$ and update $M_p$

---

## F  Experiment settings for reproducibility

### F.1  DNN Architecture

Table 10 is the detailed architecture for both AlexNet and ResNet that employed in all the experiments. For both AlexNet and ResNet, we used ReLU as activation functions.

| layer name | output size | weight shape | padding |
|---|---|---|---|
| Conv1 | $32 \times 32$ | $64 \times 3 \times 5 \times 5$ | 2 |
| MaxPool2d | $16 \times 16$ | $2 \times 2$ | |
| Conv2 | $16 \times 16$ | $192 \times 64 \times 5 \times 5$ | 2 |
| Maxpool2d | $8 \times 8$ | $2 \times 2$ | |
| Conv3 | $8 \times 8$ | $384 \times 192 \times 3 \times 3$ | 1 |
| Conv4 | $8 \times 8$ | $256 \times 384 \times 3 \times 3$ | 1 |
| Conv5 | $8 \times 8$ | $256 \times 256 \times 3 \times 3$ | 1 |
| MaxPool2d | $4 \times 4$ | $2 \times 2$ | |
| Linear | 10 | $10 \times 4096$ | |

| layer name | output size | weight shape | padding |
|---|---|---|---|
| Conv1 | $32 \times 32$ | $64 \times 3 \times 3 \times 3$ | 1 |
| Conv2_x | $32 \times 32$ | $\begin{bmatrix} 64 \times 64 \times 3 \times 3 \\ 64 \times 64 \times 3 \times 3 \end{bmatrix} \times 2$ | 1 |
| Conv3_x | $16 \times 16$ | $\begin{bmatrix} 128 \times 128 \times 3 \times 3 \\ 128 \times 128 \times 3 \times 3 \end{bmatrix} \times 2$ | 1 |
| Conv4_x | $8 \times 8$ | $\begin{bmatrix} 256 \times 256 \times 3 \times 3 \\ 256 \times 256 \times 3 \times 3 \end{bmatrix} \times 2$ | 1 |
| Conv5_x | $4 \times 4$ | $\begin{bmatrix} 512 \times 512 \times 3 \times 3 \\ 512 \times 512 \times 3 \times 3 \end{bmatrix} \times 2$ | 1 |
| Average pool | $1 \times 1$ | $4 \times 4$ | |
| Linear | 10 | $10 \times 512$ | |

Table 10: (Left:) Modified AlexNet architecture. (Right): Modified ResNet architecture

### F.2  Training Parameters in the DNN models

Table 11 shows the hyperparameters used in the AlexNet and ResNet in all the experiments, unless stated wise.

| Hyper-parameter | AlexNet | ResNet |
|---|---|---|
| Optimization method | SGD | SGD |
| Momentum | 0.9 | 0.9 |
| Learning rate | 0.01, 0.001, 0.0001† | 0.01, 0.001, 0.0001† |
| Batch size | 64 | 64 |
| Epochs | 200 | 200 |
| Learning rate decay | 0.001 at epoch 100 and 0.0001 at epoch 150 | 0.001 at epoch 100 and 0.0001 at epoch 150 |
| Weight Initialization | [15] | [15] |
| Passport Layers | Conv3,4,5 | Conv5_x |

Table 11: Training parameters for AlexNet and ResNet, respectively († the learning rate is scheduled as 0.01, 0.001 and 0.0001 between epochs [1-100], [101-150] and [151-200] respectively).

## Footnotes

[3]In this work, traditional hidden layer parameters are considered as public parameters.

[4]A rigorous proof of this argument is under investigation and will be reported elsewhere.

[5]Sergey Ioffe, Christian Szegedy, "Batch Normalization: Accelerating Deep Network Training by Reducing Internal Covariate Shift", ICML2015, pp. 448-456.

[6]Yuxin Wu, Kaiming He, "Group Normalization", ECCV2018, pp. 3-19.