[Reviews · NeurIPS 2019]

Reviewer 1



Establishing ownership of a DNN is an important problem especially given the resources and IP involved in training accurate models. The paper identifies gaps in previous efforts based on watermarking, and proposes a new method that is robust to attacks. Overall I liked the paper. It explains the key observation driving the design, and then discusses a few ways for embedding passports during training that create a dependence between the models performance and the passport input. I do have the following questions/concerns. 1) In definition 1, is D_t a pre-defined dataset i.e. known at training time? If it is not known at training time, it is unclear how M_t is computed. And if it is, is it also known to the attacker? 2) Is epsilon_f also known during training? If it is not, how does training guarantee Proposition 2, II? It appears that epsilon_f is set after training based on model performance (line 273), which seems very ad hoc. 3) The experiment to evaluate robustness to persistent reverse engineering attacks (where the adversary is assumed to have access to the training dataset) is not entirely satisfactory. The paper only explores one way of reverse engineering passports i.e. freezing training weights. There may be other ways of training e.g. freezing weights, maximizing distance from original passport, and minimizing accuracy loss. More broadly, is there an alternative way of show robustness without constraining what the adversary can do.

Reviewer 2



- Originality The method is new and differs from previous contributions. The related work is adequately cited. - Quality For most of the submission: it is technically sound; the claims are supported by theoretical analysis and experimental results; a complete piece of work; the authors are careful about evaluating the strengths and weakness of the work. The part for the resilience against ambiguity attacks is a little bit weak to me. - Clarity The paper is clearly written and well organized. - Significance The paper address an interesting task overall and present a new idea. Question about resilience to ambiguity attacks - Different from the existing watermarking methods which do not need any training/test data in verification, this method uses test performance as a metric to verify ownership — requires dataset in verification. To be fair, if anyone wants to forge watermark and is informed about the verification method (on passport layer) and verification data. They could easily learn qualified passport using same objective with the weights fixed. If they're not informed how the verification works, surely it's hard to construct ambiguity attack, but this also holds true for any other watermarking method. Together with the increasing complexity of both training and verification process, it's questionable whether people would use this idea in practice.

Reviewer 3



The paper: - shows an important weakness of the current watermarking methods, namely the fact that they are prone to ambiuity attacks, - offers an analysis of the issue investigating the requirements that have to be fullfiled by any method that should withstand such attacks, - proposes such a method based on "passport layers" which are appended after convolutions. Overall the paper is well structured and the method is explained with enough detail to probably allow reimplementation. The text is clear enough with the exception of the experiments section, which would require some additional attention from the authors. Details follow below. Concerning the method I would be interested in seing how much does the performance (accuracy) suffer because of including the passports (no passports vs. the V1 setting) and because of the multi-task setting (V2/3 vs V1). In general a comparison of the three proposed settings V1, V2, V3 is missing from the experiments/discussion. Specific comments to the experiments follow: - It is not clear whether the experiments use V1, V2, or V3? - It is not entirely clear what the Table 2 shows. I guess the numbers in parentheses are the accuracies either on the source task or after fine-tuning on the target task, and the numbers in front of the parentheses is the fraction of cases when the signature withstood the fine-tuning. Either the table headers or the legend should be improved. (Also please make the left and right tables symmetric in how the numbers are shown - with or without the "%" sign.) - In Fig. 4 legend, please specify the performance metric (accuracy?) instead of writing "DNN performances". - Consider reformulating sentences in the "Experiment results" section to make understanding the experiments easier. Especially paragraph on fine-tuning (L245-53), or sentences like "In this experiment..." (L255). Sometimes one has to search for the meaning as in the sentence "This type of weight pruning..." (L256) where it is not clear which special kind of weight pruning (if any) is refferred to. In subsection 4.2, it is not entirely clear what the "fake2" attack consists of, please clarify. - In Fig. 5, it would be helpful to specify what does "valid" and "orig" differ in. - Figures use too small font that makes reading them hard (especially Figs. 3 & 5). Please adapt the figures.

[Author Response · NeurIPS 2019]

We are grateful to reviewers for the constructive comments, which help to improve the quality & clarity of the paper.
Before addressing detailed comments, we summarize in Table 1 performances of the proposed methods under three ambiguity attack modes, $fake_i$ where $i = \{1, 2, 3\}$ depending on attackers' knowledge of the protection mechanism.

Figure 1: Test accuracy on CIFAR100 as suggested by R1 (i.e. try to create fake passport maximizing distance from $P$.)

| Ambiguity attack modes | Attackers have access to | Ambiguous passport construction methods | Invertibility (see Def. 1.V) | Verification scheme V1 | Verification scheme V2 | Verification scheme V3 |
|---|---|---|---|---|---|---|
| $fake_1$ | $W$ | Random passport $P_r$ | $F(P_r)$ failed by big margin | Accuracy $\downarrow$ (68% $\rightarrow$ 1%) with fake passport. | Accuracy $\downarrow$ (65% $\rightarrow$ 1%) with fake passport. | Accuracy $\downarrow$ (65% $\rightarrow$ 1%) with fake passport. |
| $fake_2$ | $W, \{D_r; D_t\}$ | Reverse engineer passport $P_e$ | $F(P_e)$ failed by moderate margin | Accuracy $\downarrow$ (68% $\rightarrow$ 30-45%) with fake passport. | Accuracy $\downarrow$ (65% $\rightarrow$ 20-30%) with fake passport. | Accuracy $\downarrow$ (65% $\rightarrow$ 20-30%) with fake passport. |
| $fake_3$ | $W, \{D_r; D_t\}, \{P, S\}$ | Reverse engineer passport $\{P_e; S_e\}$ by exploiting original passport P & sign string S | if $S_e = S$: $F(P_e)$ passed, with negligible margin* <br> if $S_e \neq S$: $F(P_e)$ failed, by moderate to big margin | See Figure 1 | See Figure 1 | See Figure 1 |

Table 1: Performances(%) of V1, V2 and V3 schemes under three ambiguity attack modes, $A$.

where $W$ are learned network weights; $D_r$, $D_t$ are the training and testing datasets; $F()$ is the fidelity evaluation
process, see Definition II in the main paper. * refer to $S$ encodes ownership signature, which resolves the ambiguity.

In summary, when ambiguous passports are forged and used (*e.g. forge passport/watermark with the knowledge of*
*verification method - R2 (see $fake_2$)*), Table 1 shows that all the corresponding network performances are deteriorated
to various extent. The ambiguous attacks are therefore defeated according to the fidelity evaluation process, $F()$. We'd
like to highlight that even under the most adversary condition i.e. $fake_3$ as suggested by R1, attackers are unable to
change scale signs (which encode ownership information as detailed in supplementary Table 8) without compromising
network performances. For example, with 10% and 50% of scale sign changes, the CIFAR100 classification accuracy
drops about 5% and 50%, respectively. In case that the sign remain unchanged, network ownership can be easily verified
by the pre-defined string of signs. Also, Table 1 shows that attackers are unable to exploit $D_t$ to forge ambiguous
passports (R2). We will include above results to the final draft.

Table 2 summarizes network complexity for various schemes. We believe it is the complexity and time cost during the
inferencing stage that is to be minimized, since network inferences are to be performed frequently by end users. While
extra costs at the training and verification stages, on the other hand, are not prohibitive since they are performed by
network owners, with the motivation to protect network ownerships.

| | V1 | V2 | V3 |
|---|---|---|---|
| Training | - Passport layers added <br> - Passports needed <br> - 15-30% more training time | - Passport layers added <br> - Passports needed <br> - 100-125% more training time | - Passport layers added <br> - Passports needed <br> - Trigger set needed <br> - 100-150% more training time |
| Inferencing | - Passport layers & passports needed <br> - 10% more inferencing time | - Passport layers & passport NOT needed <br> NO extra time incurred | - Passport layers & passport NOT needed <br> NO extra time incurred |
| Verification | - NO separate verification needed | - Passport layers & passports needed | - Trigger set needed (black-box verification) <br> - Passport layers & passports needed (white-box verification) |

Table 2: Summary of network complexity for V1, V2 and V3 schemes.

R1: $M_t$ is the network performance tested against $D_t$. The threshold $\epsilon_f$ is both datasets and network dependent, and
has to be set empirically by network owners, to differentiate the genuine from fake passports. Theoretical analysis of
either the threshold or its bounds might be a topic for future research.

R1: Evaluation of the cost of larger models. **Ans:** We tested a Resnet50 and its training time increases 10%, 182% and
191% respectively for V1.V2,V3 schemes. This increase is consistent with smaller models i.e. Alexnet and Resnet18.

R1: Other approaches to establishing ownership e.g. hosting models in trusted execution environments such as SGX
enclaves? **Ans:** SGX enclaves is to ensure trusted execution of models without being tampered, while the proposed
method is to protect the model from plagiarism (e.g. by a former staff who establish a new startup business with
resources stolen from network owners).

R3: Fig 4-5 not clear. **Ans:** DNN performance is test accuracy. "valid" is test accuracy using valid passports and "orig"
is baseline (unprotected model) test accuracy, we will correct "orig" to "baseline" instead.

R3: Performance (accuracy) suffer. **Ans:** Table 2 in the main paper shows the drop in test accuracy is no more than
1.5% for V1, V2, V3 compared with network without embedding any watermarks or passports.

R3: experiments uses V1, V2, V3. **Ans:** Results for all three schemes are presented in Table 2 from the main paper. We
will also add detailed comparison as outlined by Table 1 & 2 above.

R3: We will revise the final submission as requested e.g. table headers/legends and other improvements. Thanks.

[Meta-Review · NeurIPS 2019]

There is long discussion regarding the ambiguity attacks. As acknowledged by reviewers, most concerns have been addressed by the rebuttal. I would recommend acceptance.